# Directed Mutagenesis for Arginine Substitution of a *Phaseolus acutifolius* Recombinant Lectin Disrupts Its Cytotoxic Activity

**DOI:** 10.3390/ijms252413258

**Published:** 2024-12-10

**Authors:** Dania Martínez-Alarcón, José Luis Castro-Guillén, Elaine Fitches, John A. Gatehouse, Stefan Przyborski, Ulisses Moreno-Celis, Alejandro Blanco-Labra, Teresa García-Gasca

**Affiliations:** 1Centro de Investigación y de Estudios Avanzados Unidad Irapuato, Departamento de Biotecnología y Bioquímica, Irapuato 36821, Guanajuato, Mexico; dania.martinez.alarcon@gmail.com (D.M.-A.); alejandroblancolabra@gmail.com (A.B.-L.); 2Tecnológico Nacional de México/Instituto Tecnológico Superior de Irapuato (ITESI), Km. 12.5, Carretera Irapuato-Silao, El Copal, Irapuato 36821, Guanajuato, Mexico; jose.cg1@irapuato.tecnm.mx; 3Department of Biosciences, Durham University, Durham DH1 3LE, UK; e.c.fitches@durham.ac.uk (E.F.); stefan.przyborski@durham.ac.uk (S.P.); 4Facultad de Ciencias Naturales, Universidad Autónoma de Querétaro, Querétaro 76230, Querétaro, Mexico; ulisses.moreno@uaq.mx

**Keywords:** recombinant lectins, tepary bean, branched *N*-glycans, cancer

## Abstract

Recently, we reported that a recombinant Tepary bean (*Phaseolus acutifolius*) lectin (rTBL-1) induces apoptosis in colon cancer cell lines and that cytotoxicity was related to differential recognition of β1-6 branched *N*-glycans. Sequencing analysis and resolution of the rTBL-1 3D structure suggest that glycan specificity could be strongly influenced by two arginine residues, R103 and R130, located in the carbohydrate binding pocket. The aim of this work was to determine the contribution of these residues towards cytotoxic activity. Two rTBL-1 mutants were produced in *Pichia pastoris*, biochemically characterized, and cytotoxic effects were evaluated on human colorectal cancer cells (HT-29). Substitution of either of the arginine residues with glutamines resulted in significant reductions in cytotoxic activity, with losses of 1.5 and 3 times for R103 and R130, respectively. Docking analysis showed that the mutations decreased lectin affinity binding to some Epidermal Growth Factor Receptor (EGFR)-related *N*-glycans. Together, these findings confirm that both of the selected arginine residues (R103 and R130) play a key role in the recognition of tumor cell glycoconjugates by rTBL-1.

## 1. Introduction

Anticancer lectins are a specialized group of proteins that are able to recognize and interact with specific carbohydrate structures of cancer cells [1]. By specifically targeting saccharide structures, these lectins can disrupt crucial cellular processes, interfere with signaling pathways, and ultimately exert cytotoxic and antiproliferative effects on cancer cells. As a consequence, these proteins have gained considerable attention in cancer research due to their potential applications in cancer therapies [2]. One notable example of an anticancer lectin is the Phytohemagglutinin (PHA) derived from kidney bean (*Phaseolus vulgaris*) seeds. PHA has demonstrated cytotoxic effects on cancer cells, leading to cell death and reduced tumor growth [3]. PHA’s anticancer effect stems from its ability to recognize and bind to specific carbohydrate molecules displayed on the surface of cancer cells. By binding to these structures, PHA can interfere with vital cellular processes, disrupt signaling pathways, and induce various anti-tumor effects, including the induction of apoptosis, inhibition of cell proliferation, modulation of immune responses, and inhibition of angiogenesis [4,5,6,7].

In 2012, we reported that a group of PHA-homologous native lectins derived from Tepary bean (*Phaseolus acutifolius*) exhibited cytotoxic effects on various types of cancer cell lines [8] by inducing apoptosis and cell cycle arrest [9]. It was found that the Tepary Bean Lectin Fraction (TBLF) contained at least two cytotoxic lectins that were characterized at molecular and bioactive levels [10,11]; furthermore, when administered orally, TBLF exhibited the ability to inhibit early colon tumorigenesis in rats [12]. As Tepary Bean Lectin-1 was identified as the main lectin responsible for this cytotoxic effect, we subsequently produced it as a recombinant protein by bench-top fermentation of transformed *Pichia pastoris* and determined its three-dimensional structure [13,14]. The recombinant lectin-1 (rTBL-1) was found to recognize β1-6 branched *N*-glycans, which are overrepresented in several cancer types, including colon cancer. Recognition of β1-6 branched *N*-glycans does not appear to be affected by elongation with more units as sialic acid [13]. We also found that rTBL-1 retains its cytotoxic apoptosis-inducing effect on colon cancer cells via influencing epidermal growth factor cell receptor (EGFR) signaling pathways and principally by causing partial receptor degradation and activation of p38 MAPK signaling [15].

The structural characterization of rTBL-1 enabled us to compare its putative binding pocket and residues responsible for binding to the trisaccharide unit Gal(β1-4)GlcNAc(β1-2)Man with that previously reported for PHA [16]. We identified that, unlike other closely related legume agglutinins with anticancer activity, rTBL-1 has an arginine residue instead of lysine at position 103, which could be involved in the formation of hydrogen bonding to carbohydrates. Additionally, we identified the presence of an additional arginine within the binding pocket and adjacent to other residues involved in the recognition of carbohydrates reported for PHA. Therefore, as a highly conserved arginine residue is crucial for carbohydrate recognition in other families of lectins, such as the siglectins (sialic acid immunoglobilin-superfamily lectins) [17,18], we hypothesize that the sequence discrepancy in rTBL-1 could explain its cytotoxic effect as compared to the legume leucoagglutinins. To investigate this further, we have carried out a conservative substitution of two arginine residues with the uncharged amino acid glutamine in order to evaluate the impact of these modifications on the cytotoxic effect of rTBL-1 lectin.

## 2. Results and Discussions

### 2.1. Production of rTBL-1 Mutants

Multiple alignments of leucoagglutinin sequences, along with the findings of Nagae et al. [16], revealed that all sequences have identical carbohydrate binding pockets (CBP) (Figure 1A, blue triangles), except *P. acutifolius* and rTBL-1, where an arginine (R) instead of lysine (K) is located at position 103. This change is particularly interesting because, despite the similarity between these amino acids, conservative substitutions such as replacing R with K in other lectins, such as the animal I-type lectins (siglectins), have shown a conserved arginine residue to be essential for sialic-binding [17,19]. Sialic acid is particularly abundant in tumor cells where abnormal sialylation is considered as a hallmark of cancer [20], so it has been studied for cancer-involved membrane receptors such as EGFR [21].

An additional arginine residue located at position 130 (R130) in TBL-1 differs from the presence of a lysine in the *P. vulgaris* lectin or tryptophan in *P. costaricensis* or *P coccineus* lectins. R130 does not appear to have direct interaction with the ligand but is located adjacent to two of the residues involved in carbohydrate recognition and also for metal-bonding cations. These bonds are essential for interactions between monomers and, therefore, for the recognition of complex carbohydrates. It is thus possible that R130 may contribute to enhancing the cytotoxic effects of TBL-1. It was also found that the sequon, unique consensus sequence for *N*-glycosylation (N-X-S/T, where X represents any amino acid except P) [22] was located between residues 13 and 15 (N-E-T).

Two rTBL-1 mutants were generated by substituting either R-103 or R-130 within the CBP with a non-charged glutamine (Q) residue. The resulting proteins were designated as R103Q and R130Q mutant lectins, respectively. Site-directed mutagenesis was performed on the original coding sequence of rTBL-1 by inverse amplification of the pGAPZαB-rTBL-1 plasmid expression vector [13] to obtain amplicons of approximately 3000 bp (Figure 1B). Whilst amplification parameters were tested at two different temperatures (62 and 59 °C), viable products were only obtained at 59 °C (Figure 1B).

The vectors were gel purified, re-ligated, and subsequently cloned into Top10 *Escherichia coli* cells. Sequence verified plasmid DNAs were linearized and integrated into the GAPDH locus of *P. pastoris* through homologous recombination. Four colonies of each mutant were selected for expression screening via western analysis of supernatants derived from shake flask cultures. As shown in Figure 1C, all selected colonies secreted a recombinant protein of approximately 32 kDa consistent with the predicted mass for the mutants. The presence of an additional immunoreactive protein (ca. 25 kDa) observed in 6 of the 8 culture supernatants suggests degradation at the *N*-terminus of the mutants, since the His tag is at the C-terminus of the protein. Based on the performance of the small-scale cultures, colony number #1 for each mutant was selected to scale up production in a 7 L bench top fermenter.

The electrophoretic profile of the culture media obtained for both fermentations was very similar to the one previously reported for the rTBL-1 fermentation, where the recombinant product is presented as the most predominant band. The final yields per liter of culture supernatant following bench-top fermentation were 205 mg and 184 mg, respectively, for R103Q and R130Q. Proteins were purified by nickel affinity chromatography, and fractions were collected after elution with 200 mM imidazole. Analysis of purified proteins by SDS-PAGE electrophoresis showed the presence of a single band of ~30 kDa in Coomassie-stained gels (Figure 1D,E).

### 2.2. R103Q and R130Q Biochemical Characterization

rTBL-1 and the purified mutants were all separated as proteins of approx. 30 kDa under denaturing SDS-PAGE gel electrophoresis in line with predicted masses for the recombinant lectins (Figure 2A). However, it is well-known that rTBL-1 folds into a homo-tetramer [13]. To assess the oligomeric forms of R103Q and R130Q mutant lectins, their native molecular weights were determined using gel filtration chromatography using a high-resolution size exclusion column (ENrich™ SEC650, BioRad. Hercules, CA, USA). The elution profiles exhibited a single peak at approximately 120 kDa, indicating that both mutants are able to conform homo-tetramers in solution and that the K to Q substitutions did not disrupt their ability to oligomerize.

A single *N*-glycosylation site near their *N*-terminal region was found for rTBL-1; this sequence leads to the addition of an *N*-glycosidic antenna of approximately ~2.48 kDa on asparagine-13. All recombinant proteins were treated with various combinations of glycosidases in order to compare the type and estimated size of attached glycosidic antennas. Subsequently, the samples were subjected to SDS-PAGE, and the gels were stained using the Schiff-PASS technique, enabling the identification of protein-bound carbohydrates (Figure 2B).

Both R103Q and R130Q mutants showed similar molecular weights (Figure 2A). As rTBL-1 and the two mutants showed no changes in the protein sequon, an analysis of the glyosidic antennas was performed using several glycosidases. Upon treatment with *N*-glycosidase-F, both mutants and rTBL-1 exhibited a reduction in size of between 1.84 and 2.48 kDa, accompanied by the complete carbohydrate removal. No changes in intensity or molecular size were observed after treatment with *O*-glycosidases indicating that rTBL-1 and the mutants contain only *N*-linked glycans. Nano-LC-MS/MS analysis depending on standard data of tryptic peptides, was used to confirm the position of the carbohydrate antennas from samples treated with deglycosylating enzymes. This analysis confirmed the presence of an *N*-deamidated asparagine in the NETN peptide, which corresponds to positions 13–16 of the proteins. Deamination is known to be a byproduct of PNGaseF treatments by removing *N*-linked sugars.

**Figure 2 ijms-25-13258-f002:**
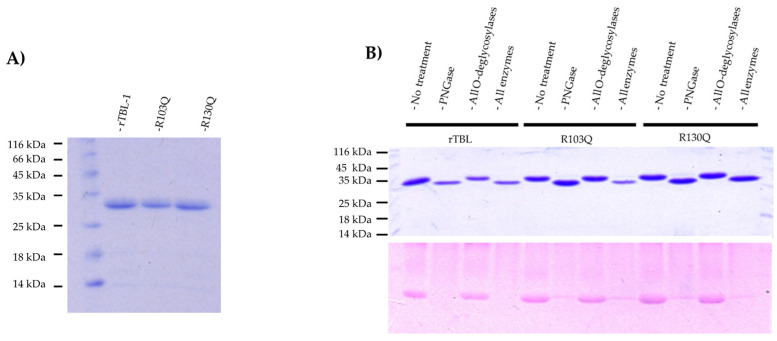
Characterization of the mutants. (**A**) Molecular size comparison between rTBL-1 and the mutants. SDS-PAGE of the electrophoretic profiles of rTBL-1, R103Q, and R130Q. Lane 1, ladder; lane 2, 5 µg of rTBL; lane 3, 5 µg of R103Q; lane 4, 5 µg of R130Q. (**B**) Analysis of glycosylation. Top panel shows a coomassie R-250-stained SDS-PAGE. *N*-Glycosidase F (PGNaseF), all *O*-glycosidases (β1,4-Galactosidase, endo-α-*N*-acetylgalactosaminidase, α2-3,6,8,9-Neuraminidase, and β-*N*-Acetylglucosaminidase), all *N*- and *O*-glycosidases (*N*-Glycosidase F, β1,4-Galactosidase, endo-α-*N*-acetylgalactosaminidase, α2-3,6,8,9-Neuraminidase, and β-*N*-Acetylglucosaminidase).

### 2.3. Cytotoxic Effect of R103Q and R130Q

The cytotoxicity of rTBL-1 and its mutants was assessed on HT-29 colorectal cancer cells through dose–response curves using concentrations ranging from 0 to 50 μg/mL (Figure 3). The percentage of cell survival was normalized against the negative PBS-treated control treatment. The results show concentration-dependent responses for all proteins. However, rTBL-1 showed significantly greater cytotoxic effects as compared to either of the mutants, as evidenced by significant differences (*p* < 0.05) in lethal concentrations (LC_50_ values) derived from dose–response assays. LC_50_ values were 2.37 μg/mL (y = −16 − 317x + 88.654) for rTBL-1 as compared to 4.94 μg/mL (y = −12.702x + 112.59) and 7.22 μg/mL (y = −8.6725x + 112.7) for R103Q and R130Q, respectively.

These results show that each arginine substitution plays a role in exerting the cytotoxic effects of rTBL-1. R130Q exhibited the greatest inhibition of cytotoxicity, being 3 times less cytotoxic as compared to rTBL-1 and 1.5 times lower than R103Q. No statistical differences (*p* > 0.05) were observed for all tested concentrations of R130 as compared to control cells. For R103, significant differences (*p* < 0.05) were observed for the 5 to 50 μg/mL treatments as compared to the control cells. These results highlight the differential role of arginine residues 103 or 130 in the lectin’s carbohydrate binding activity.

**Figure 3 ijms-25-13258-f003:**
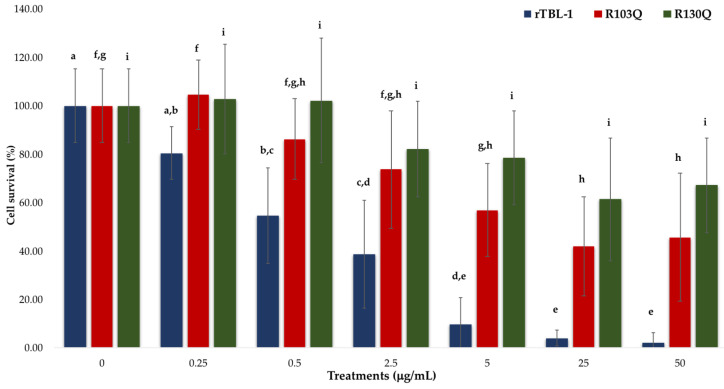
Cytotoxic effect of *r*TBL-1 and its mutants on HT-29 colon cancer cells. Cells were treated with rTBL-1, R103Q, or R130Q (0, 0.25, 0.5, 2.5, 5, 25, and 50 μg/mL) for 24 h and cell survival percentages were calculated from hemocytometer counts. Small letters show significant differences for each separate treatment at different concentrations (Tukey *p* < 0.05), where differences between treatments for rTBL-1 were pointed out with letters a–e, for R103Q with letters f–h, and for R130Q with letter i.

### 2.4. Docking Analysis

In order to know more about the participation of R103 and R130 residues, a docking analysis was performed using the rTBL-1 sequence (PDB ID: 6tt9); this sequence has one extra alanine beside the first serine of the rTBL-1 sequence [14]. This change did not affect the tridimensional structure, but it was necessary to go through the numbering one position, changing R103Q to R104Q and R130Q to R131Q only for the docking analysis. On the other hand, all docking procedures were performed using the LZERD web server according to its specifications (https://lzerd.kiharalab.org accessed on 28 January 2023) [23].

#### 2.4.1. Docking Analysis with Experimentally Recognized *N*-glycans

Previously we have shown that rTBL-1 recognized experimental complex *N*-glycans from the Mammalian Glycan Array (GA) 5.4; therefore, a docking analysis was performed considering only those whose relative signal was above 40% [13]. Figure 4 shows the interaction between rTBL-1 and experimental *N*-glycans, where GA-01, GA-02, GA-03, GA-04, GA-06, GA-07, and GA-09 exhibited a docking interaction in the upper zone between monomers 1 and 2 of tetrameric rTBL-1 located near the CBP, confirming recognition and suitable interaction positions of the experimentally recognized *N*-glycans. Only those complex glycans having four sialylated branches (GA-08) or highly elongated branches (GA-05) interacted directly in the CBP with a large part of the glycan structure. These results provide the first two selection criteria for those *N*-glycans that had direct and considerable interactions with the lectin CBP, namely the presence of at least four-ended sialylated branches and/or elongated branches.

#### 2.4.2. Docking Analysis with EGFR-Related *N*-glycans

EGFR is an important glycoprotein for cancer biology, being highly expressed and considered as a proteomic marker in different types of cancers [24,25,26]. EGFR-specific *N*-glycosylations are correlated with disease state/progression of colorectal cancer [27], and, for this reason, EGFR-related *N*-glycans were considered in this work for docking analysis using the structures of rTBL-1 and the in silico mutated versions R104Q and R131Q.

**Figure 4 ijms-25-13258-f004:**
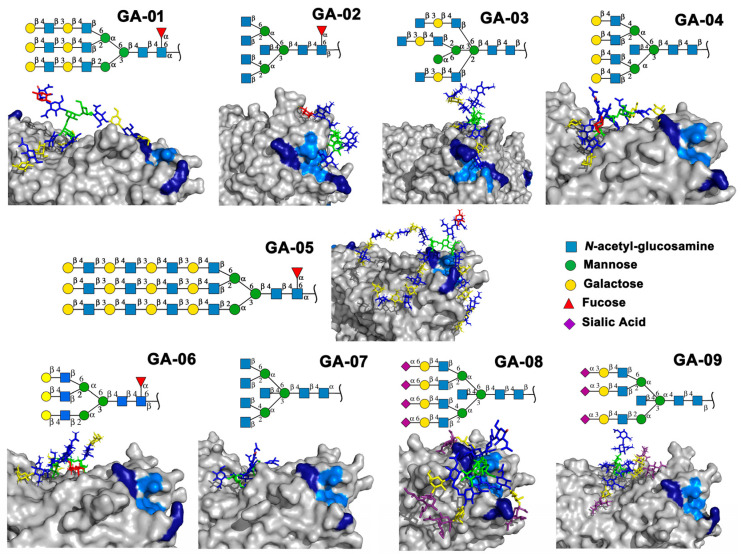
Molecular docking of the experimental *N*-glycans from the Mammalian Glycan Array 5.4 with rTBL-1 structures. The graphical 2D representation of *N*-glycans is shown according to the Symbol Nomenclature for Glycans (SNFG) [28]. The *N*-glycans from the mammalian glycan array “Version 5.4” are labeled as GA-##, where ## is the number described by Martínez-Alarcón et al. (2020) [13] and is related to the signal’s relative intensity of lectin recognition. Docking results are shown as topological representations of interactions at the same surface location between the *N*-glycans with the recombinant lectins: rTBL-1, R104Q, and with the R131Q lectin. The *N*-glycan in topological representations is presented as sticks colored according to the SNFG (as indicated by geometric codes), the protein surface in gray, highlighting the CBP in light blue and the arginine residues in dark blue. GA, glycan array.

Complexity in the structure of *N*-glycans was also considered to select the EGFR-related glycans for the docking analysis, since rTBL-1 previously showed high affinity for complex glycans [13]. Furthermore, the protein–protein interaction of rTBL-1 with the EGFR has been observed [15]. On the other hand, due to the possible positions of its flexible structure, it was necessary to use several EGFR-related *N*-glycan conformers generated on the GLYCAM web server [29] in each docking analysis. This allowed us to screen a wide variety of docking results and select those within the CBP, and specifically those whose interactions were related to the mutated residues. Thirty-two of the 52 EGFR-related *N*-glycans (GlyConnect-https://glyconnect.expasy.org accessed on 8 February 2023) [30] were used for docking analysis with rTBL-1.

Only eight *N*-glycans were not recognized by rTBL-1 (considering that the resulting interactions were located between monomers [11]), and 24 EGFR-related *N*-glycans had reliable docking positions, including six interacting directly with CBP (Figure 5). Interestingly, when the reported molecular mass of each selected *N*-glycan (GlyConnect platform, https://glyconnect.expasy.org/ accessed on 8 February 2023) [30] was compared, a molecular mass-lectin recognition correlation was found (Appendix A). The unrecognized *N*-glycans (those *N*-glycans with unsuitable docking interactions) had an average mass of 1.77 ± 0.33 kDa and included mainly complex two-branched glycans and high-mannose *N*-glycans. The *N*-glycans whose docking interaction was appropriate but without considerable interactions with CBP had 2.26 ± 0.25 kDa as average mass. Some of these *N*-glycans have three branches, two of which are located directly on the mannose (α1-6) of the *N*-acetyl-glucosamine-trimannoside core. Most of these *N*-glycans were fucosylated. Finally, the EGFR-related *N*-glycans that had multiple interaction points with the CBP of rTBL-1 showed an average mass of 4.05 ± 0.25 kDa, having four branches containing sialic acid (*N*-acetylneuraminic acid [NeuAc(α2-3)]) at the end of each branch and presenting a *N*-glycan core fucosylation at the innermost *N*-acetylglucosamine residue of the chitobiose (GlcNAc_2_) core. An interesting feature of these *N*-glycans is that their structure was formed by four sialylated antennae and a core-fucosylated structure.

It is worth mentioning that all those EGFR-related *N*-glycans whose interaction with Tepary bean lectin-related structures (rTBL-1, R104Q, and R131Q) have multiple contact points on the CBP region, especially for core-fucosylated and highly sialylated *N*-glycans. This fact is important since core fucosylation has been associated with a tumorigenic phenotype, and direct inhibition is able to inhibit cellular signaling for cell proliferation, migration, and tumor growth [31,32]. On the other hand, aberrant sialylation in cancer cells plays an important role in tumor progression, favoring the evasion of cell death and promoting metastasis, immune evasion, and drug resistance. Aberrant sialylation, specifically α2-3- and α2-6-linked sialic acids, has been reported in highly metastatic and drug-resistant colorectal cancer and other tumor cells [33,34,35,36,37,38]. Highly sialylated *N*-glycans of EGFR have also been shown to disrupt ligand interactions by avoiding dimerization and subsequent phosphorylation [39], thereby protecting tumors against drug-induced cell death [40]; therefore, sialylation is an important target against cancer progression [41]. In consequence, the high affinity of rTBL-1 for highly sialylated EGFR-related *N*-glycans, especially those having α2-3-linked sialic acids, suggest that its interaction directly promotes apoptosis of colon cancer cells, and this preference could explain the anticancer effect of this lectin and potential uses for the treatment of other types of cancer.

#### 2.4.3. The Role of R104 and R131 Residues in the Affinity rTBL-1 Toward EGFR-Related *N*-glycans by Docking Analysis

To determine the effect of punctual mutations on the lectin’s recognition of EGFR-related *N*-glycans, the high-molecular-mass *N*-glycans that interacted directly with a large part of their structure on the lectin CBP (forming a preliminary docking process using the structures shown in Figure 5) were chosen for docking analysis with in silico-mutated lectin structures (R104Q and R131Q). Considering that all *N*-glycans have a Man_3_GlcNAc_2_ core, in this work the branches or “antennae” of complex *N*-glycans were numbered according to their relative position with respect to the mannose residue to which they are attached, either to Man(α1-6) or Man(α1-3) to simplify the nomenclature of the EGFR-related *N*-glycans (Appendix A). For instance, branch 1 corresponded to the antennae attached to carbon 6 of Man(α1-6) of the Man_3_GlcNAc_2_ core, and branch 2 corresponded to the antennae attached to carbon 2 of the same mannose residue. In the same way, branch 3 corresponded to carbon 4 of Man(α1-3) of the Man_3_GlcNAc_2_ core, and branch 4 corresponded to the antennae attached to carbon 2 of the same mannose residue.

Some of the EGFR-related *N*-glycans have elongated antennae. Generally, branches of complex *N*-glycans are composed of NeuAc(α2-3)Gal(β1-4)GlcNAc(β1). However, in elongated antennae, two additional monosaccharides are present, which are *N*-acetylglucosamine with a glycosidic bond β1-3 [GlcNAc(β1-3)] and galactose attached to the *N*-acetylneuraminic acid (sialic acid) [NeuAc(α2-3)] found at the end of the branch, as follows: NeuAc(α2-3)Gal(β1-4)GlcNAc(β1-3)Gal(β1-4)GlcNAc(β1).

Figure 6 shows the topological representation of the docking interactions of the selected EGFR-related *N*-glycans and the three lectin structures: rTBL-1, R104Q, or R131Q. The lectin structure was colored as a gray surface, and the CBP zone was highlighted as a light-blue zone, with residues R104 and R131 as dark-blue zones and mutated residues (Q104 and Q131) as dark-red surfaces, in order to compare the different ways in which the glycans interacted with the CBP of the lectins. When *N*-glycans interacted with rTBL-1, all except *N*-glycan 876 had a considerable interaction area on the CBP surface. Then, when comparing the docking results of the R104Q lectin with the EGFR-related *N*-glycans, *N*-glycans such as 876 and 2132 had no interaction surface in the CBP. In the case of R131Q lectin, the interaction positions were slightly changed compared to those of the rTBL-1 lectin, except for glycan 876, which had a very different location.

When comparing the 3D-interaction maps of all docking interactions (Figure 7), where protein-interacting residues are highlighted as orange sticks with their respective surface and chain position number, it is noteworthy that in the interactions of the *N*-glycans with the lectin rTBL-1, the R131 residue participates in most of them (1944, 1658, 2132, and 3414), and the R104 residue only in the interactions between rTBL-1 and 2132, 2608, and 3414 glycans. Both residues interact by a hydrogen bond (Appendix A) and this could explain how the change of arginine to glutamine allowed hydrogen bond interactions by the amino group of the glutamine residue. Additionally, the Q104 residue participated in most of the interactions with *N*-glycans, such as 876, 1944, 1658, 2608, and 3414; in the R104Q lectin—2608 *N*-glycan interaction, it was possible to observe the participation of the R131 residue. Finally, the mutated Q131 residue participated only in the interactions of 1944, 1658, and 2608 *N*-glycans, having the additional participation of the R104 residue in the case of the interactions with 1658 and 2608 *N*-glycans.

**Figure 6 ijms-25-13258-f006:**
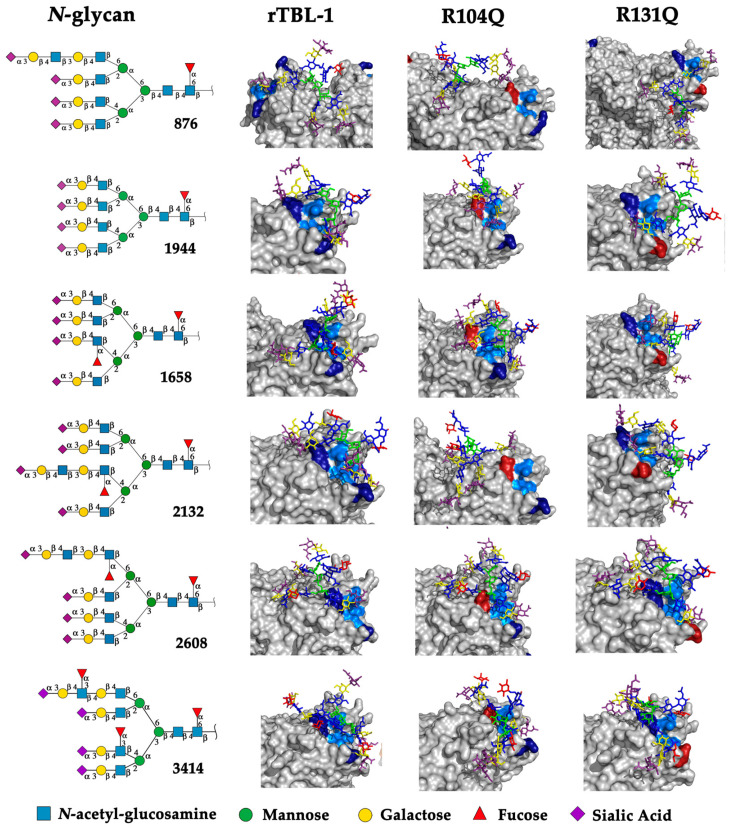
Molecular docking results for EGFR-related *N*-glycans with rTBL-1 and mutants R104Q and R131Q. For each *N*-glycan a graphical 2D representation according to SNFG [28] is shown. Docking results are shown as topological representations of interactions at the same surface location between the *N*-glycans, and the recombinant lectins: rTBL-1, R104Q, and R131Q. The *N*-glycan in topological representations is presented as sticks colored according to the SNFG, the protein surface in gray, highlighting the CBP in light blue, the mutated residues in red, and the original arginine residues in dark blue.

At this point, it is remarkable that the arginine to glutamine mutation (R → Q) allows, in most cases, the interaction of the lectin with *N*-glycans. Nevertheless, the R104Q mutation completely changed the interaction profile with the EGFR-related 2132 *N*-glycan, which has an elongated antenna with a second fucosylation at branch 3. This change is evident in the respective docking analysis, as there were no interactions in the CBP zone. This may be an indication that R104 plays a crucial role in the specific recognition of the lectin to certain *N*-glycan structures, and although the mutated residues may interact with *N*-glycans in most cases, they affect the three-dimensional configuration of the CBP and its surrounding regions and ultimately influence the specific recognition of the lectin and its target *N*-glycans.

**Figure 7 ijms-25-13258-f007:**
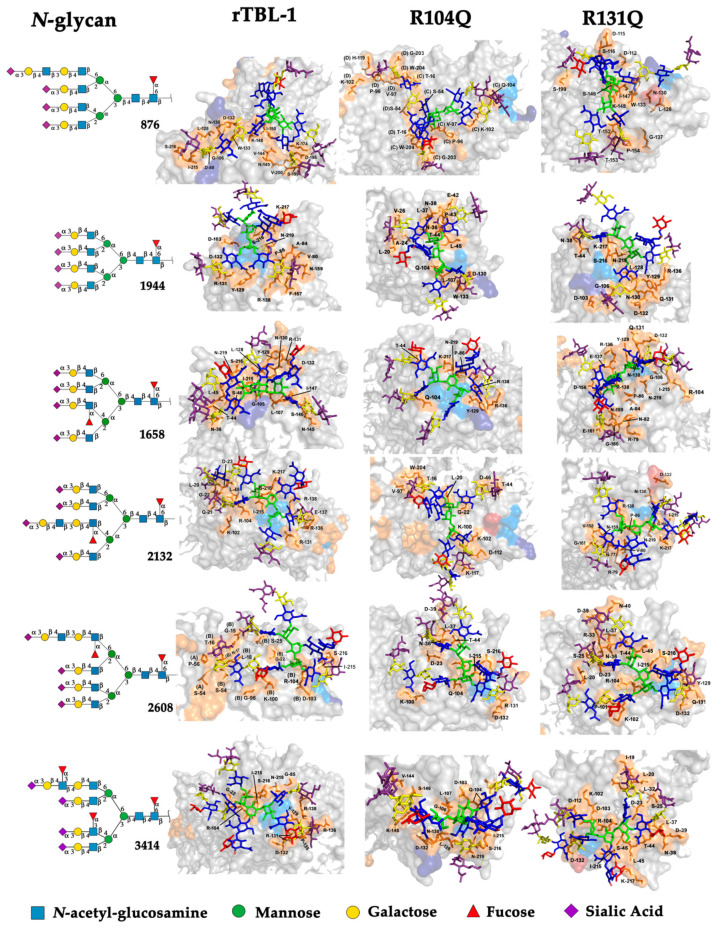
Three-dimensional maps for the interactions between EGFR-related *N*-glycans to rTBL-1, and mutants R104Q and R131Q. For each *N*-glycan, a graphical 2D representation according to SNFG [28] is shown. The interaction maps are shown as topological representations of the interactions of the *N*-glycans and the recombinant lectins: rTBL-1, R104Q, and R131Q. The target *N*-glycans are shown as sticks, and the structure of *N*-glycan residues is colored according to SNFG. The docking-interacting residues of lectins are shown as orange sticks with their respective surface and relative sequence number as labels.

In order to characterize the interaction interfaces between the EGFR-related *N*-glycans and the CBP of the recombinant and mutant Tepary bean lectins, the solvation free energy gain upon formation of the interface (Δ^i^G) was determined for all docking interactions (Table 1). The Δ^i^G was calculated as a difference in solvation energies of all residues between interfacing structures, and a negative value corresponds to hydrophobic interfaces or is indicative of positive protein affinity according to the docking methodology of the PDBePISA web server [42].

An important finding regarding the role of residues R104 and R131 in the recognition of complex *N*-glycans, considering the Δ^i^G, is the appreciable loss of affinity in the recognition of EGFR-related *N*-glycan 2132 when R104 is mutated. Although the glycan is recognized and interacts with the lectin, it no longer interacts directly with the CBP and its solvation free energy upon formation of interfaces (Δ^i^G) decreases considerably, indicating a significant loss of affinity (compare the docking results of glycan 2132 (Figure 5) with the energies of Table 1). This phenomenon also occurs with the interaction between lectins with EGFR-related *N*-glycan 876, which has an elongated antenna at branch 1 without a second fucosylation, but it is not as representative since this glycan interacts with CBP with few residues with the rTBL-1. Additionally, this could also provide insight into the importance of *N*-glycan 2132 as a target molecule for tepary lectin and the effect it has on colon cancer.

Another interesting finding is the probable effect of fucosylations on the recognition of *N*-glycans by tepary bean lectins. While the *N*-glycan 876 with elongated branch 1 did not have much interaction with CBP and even lost it when the residues were mutated, the *N*-glycan 2608, which had a similar structure but with a fucosylation on elongated branch 1, did have considerable interaction with CBP, which was maintained (and even apparently enhanced in the R131Q mutant), suggesting that this fucosylation could be important to recognize this structure with an elongated branch 1 (compare Figure 5 and Table 1). Fucosylations in non-elongated branches apparently have no effect (compare interactions with 1944 and 1658 EGFR-related glycans in Figure 5); however, when a fucosylation in branch 3 is present, the values of the Δ^i^G increased apparently even more in the interactions with mutated residues, indicating an apparent improvement in affinity between the interfaces of the interaction (compare Δ^i^G values of 1944 and 1658 *N*-glycans in Table 1). In addition, the presence of a third fucosylation in N-glycans (i.e., 3414 EGFR-related *N*-glycan) also showed enhanced affinity in the case of the interaction with rTBL-1, but a considerable decrease in the interactions with R104Q and R131Q mutated lectins.

The docking results suggested that the observed loss of cytotoxicity of the mutants was related to the loss of affinity of the studied lectin-*N*-glycan interactions. In fact, each interaction depends on the glycan type suggesting a specific effect. Collectively, the results show that the interactions with *N*-glycans, such as 876, 1658, 2132, and 3414, could promote cytotoxicity since important changes in solvation free energy gain were determined for one or both mutants for each *N*-glycan. In particular, the substitution of R104 with Q104 decreased the affinity and (in some cases) the location of the interaction for three of the six tested *N*-glycans, and an important increase was only observed for one of them. On the other hand, for the substitution of R131 for Q131, an increase in affinity was observed for three *N*-glycans and a decrease for two. Both the decrease and increase in interactions compared to rTBL-1 could influence the properties of the lectin, where a decrease is primarily associated with a loss of affinity, while an increase could be linked to binding dissociation or a regulatory function.

## 3. Materials and Methods

### 3.1. Site-Directed Mutagenesis

To generate the mutants, two sets of primers containing the specific mutations were employed to amplify the recombinant plasmid pGAPZαB-rTBL-1, previously generated by Martinez-Alarcon et al. [13]. The modifications were introduced near the 5′ end of the reverse primers (underlined content). R103Q was generated using primers 126_F (5′-GGTCTTCTAGGTCTGTTCGACGGC-3′) and 126_R (5′-GCCTTGGTCTTTGGGCTTAGAGC-3′), while R130Q was generated using primers 152_F (5′-CAGAGAGCGTCATATTGGCATCG-3′) and 152_R (5′-GGGTCCCAGTCCTGGTTGTACAAG-3′). The amplified products were gel purified using a commercial BioLabs kit (Monarch DNA Gel Extraction. Hitchin, UK). Purified plasmids were subsequently ligated at 25 °C for 1 h using T4 ligase. The ligation products were transformed into *Escherichia coli* TOP10 cells (Invitrogen Life Technology, Cat. C4040-06. Waltham, MA, USA) via electroporation (resistance of 100 Ohms, capacitance of 125 μFa, and 1.8 Volts), and cells were plated on Luria–Bertani (LB) medium containing 25 μg/mL zeocin as a selection agent, followed by incubation at 37 °C. After 24 h, transformants were screened by colony PCR (using gene-specific primers or pGAPαB primers), and positive clones were verified by digestion and sequencing.

### 3.2. Transformation of Pichia pastoris

The procedure was followed based on the patent MX/a/2018/009442. Sequence-verified plasmids were digested with *Avr*II at 37 °C for 16 h, and complete linearization of the DNA was confirmed through nucleic acid electrophoresis. Subsequently, the DNA was ethanol precipitated, and the pellet was resuspended in 20 μL of nuclease-free sterile water. The concentration, determined by NanoDrop spectrophotometry (NanoDrop 2000c from Thermo Scientific; Hercules, CA, USA) and electrophoresis, was confirmed to be >5 μg/μL. This DNA material was used to transform *Pichia pastoris* protease-deficient strain SMD1168H (Invitrogen Life Technology, Cat. C18400. Waltham, MA, USA) using the Pichia EasyCompTM Transformation Kit (Thermo Fisher Scientific; Pittsburgh, PA, USA) following the manufacturer’s protocol. Transformed cells were recovered and plated on Yeast Extract-Peptone-Glycerol (YPG) medium supplemented with 50 μg/mL zeocin and incubated at 30 °C for 48 h.

Ten colonies of *P. pastoris* were selected, and each one was inoculated into 10 mL of liquid YPG medium supplemented with 25 µg/mL zeocin. The cultures were incubated at 30 °C with shaking for 48 h, after which the supernatants were collected by centrifugation at 11,800× *g* for 10 min. For SDS-PAGE electrophoresis, 25 µL aliquots of each supernatant were loaded and transferred onto nitrocellulose membranes using a semi-dry transfer system (ATTO blotter. Tokyo, Japan) at a constant voltage of 10 V for 1 h in 1X TBS (Tris-Buffered saline, Tris-HCl 15.2 mM, Tris base 46.2 mM, and NaCl 1500 mM). Ponceau red staining was performed to visualize the positions of molecular size markers, followed by blocking the membranes with 5% (*w*/*v*) defatted milk powder in 0.1% (*v*/*v*) Tween TBS for 1 h. The membranes were incubated overnight with anti-histidine monoclonal antibodies at a 1:1000 dilution in blocking solution. After 16 h, the primary antibodies were removed by several washes with blocking solution, and the membranes were further incubated for 2 h in blocking solution containing secondary anti-goat antibodies coupled to alkaline phosphatase. Finally, chemiluminescence was used to visualize the immunoreactivity.

### 3.3. Heterologous Expression and Purification of R103Q and R130q Mutants

*P. pastoris* cells expressing recombinant rTBL-1 were cultivated in a 7.5 L vessel Applikon ez-control laboratory fermenter, as described previously [13], with the exception of maintaining the pH at 5.0. After cultivation, the culture was centrifuged at 7500× *g* for 30 min at 4 °C to separate the secreted proteins from the cells. The resulting supernatant was clarified by sequential filtration using 2.7 μM and 0.7 μM glass fiber filters (Whatman. Kent, UK). Purification of the recombinant proteins from the supernatant was performed using nickel-affinity chromatography, followed by dialysis and freeze-drying, following the established protocol [14]. The protein content in the lyophilized samples was determined by SDS-PAGE gels, where the bands corresponding to intact proteins were compared to GNA (Sigma Aldrich. Burlington, MA, USA) standards through visual inspection and by capturing an image of the destained gel using a commercial flat-bed scanner. Image analysis was conducted using a custom-written software “ProQuantify”, version 1.0, supplied by Rodrigo Guerrero from Universidad Autónoma de Queretaro, Mexico.

### 3.4. Size Exclusion Chromatography

Size exclusion chromatography (SEC) was carried out using a High-Resolution ENrich™ SEC 650 column (Bio-Rad. Hercules, CA, USA) on an NGC™ chromatography system (Bio-Rad). Prior to the assay, the column was calibrated with the gel filtration standard #15119001 (Bio-Rad), following the supplier’s instructions. Protein samples at a concentration of 10 mg mL-1 were centrifuged at 8000× *g* for 30 min. The column was equilibrated with 50 mL of buffer D (20 mM MES, 100 mM NaCl, pH 6.5), and then 200 µL of the sample were injected into the system, followed by a 40 mL isocratic elution using buffer D. Fractions were monitored at 280 nm for absorbance and collected.

### 3.5. Identification of the Glycosylation Present in the Recombinant Lectin

To determine the size and composition of the glycosidic antennas on Mut126 and Mut153, the proteins underwent glycan digestion using a combination of five distinct glycosidases that target O and *N*-glycosylations. This process was performed utilizing the Glycoprotein Deglycosylation kit (EMD Millipore. Darmstadt, Germany) according to the manufacturer’s instructions.

The glycosidases used were: (i) *N*-Glycosidase F: this enzyme cleaves all *N*-linked oligosaccharides, unless their core is α1-3 fucosylated; (ii) Endo-α-*N*-acetylgalactosaminidase that separates common nucleus (Galβ1,3GalNAca) from all *O*-glycosylations (it only works when no additional sugars are adhered to the main nucleus); (iii) α2-3,6,8,9-Neuraminidase that cleaves all sialic acids that may be attached to the common core of the *O*-glycosylations; (iv) β1,4-Galactosidase that releases only non-reducing terminal galactose, bound in β1-4 to *O*-glycosylations; (v) β-*N*-Acetylglucosaminidase that cleaves all GlcNac residues bound to β-non-reducing terminal *O*-glycosylations. Following digestion, samples were subjected to SDS-PAGE and subsequently stained with Coomassie blue. Additionally, the periodic acid-Schiff (PAS) staining technique (Sigma Chemical, St. Louis, MO, USA) was employed in accordance with the prescribed procedure based on the method originally proposed by Hotchkiss [43].

### 3.6. Cytotoxic Activity

To assess the cytotoxic activity of the mutants, HT-29 human colon cancer cells (ATCC, HTB-38. Manassas, VA, USA) were employed. Initially, 10,000 cells were seeded per well in 48-well plates containing 0.5 mL of medium A, which consisted of DMEM (Sigma-Aldrich, D-5796. Burlington, MA, USA) supplemented with 10% *v*/*v* fetal bovine serum (FBS) (Sigma-Aldrich, 12003C) and 1% *v*/*v* L-glutamine-antibiotics (Sigma-Aldrich, G6784). The plates were incubated at 37 °C with 5% CO_2_ and 95% relative humidity until cells reached 80% confluence. After 48 h, the culture medium was replaced with medium B, composed of DMEM supplemented with 2% FBS *v*/*v* and 1% L-glutamine *v*/*v*, to synchronize the cells. Following an additional 24 h incubation, the medium was removed, and the cells were treated with DMEM supplemented with 0.5% BSA *w*/*v* (AlbuMax. Life Technologies, 11020021. Carlsbad, CA, USA) 1% L-glutamine *v*/*v*, and various concentrations of the respective treatments. PBS was used as the negative control, while rTBL served as the positive control. After 24 h incubation, the cells were harvested using 0.15% trypsin *w*/*v* and counted using a hemocytometer; triplicate wells were counted per treatment to determine cell numbers. Cell survival was calculated considering the negative control to be 100%. Lethal concentration 50 (LC_50_) was calculated by plotting the log_10_ of the treatment’s concentration vs. cell survival % and taking the value of the ordered pair (*y*) as 50%.

Statistical analysis was determined by comparing the mean values for each treatment using a one-way ANOVA with a Tukey post hoc analysis (*p* < 0.05) to determine which means were significantly different.

### 3.7. Docking Analysis Procedure 

EGFR-related *N*-glycans were identified using the GlyConnect platform (https://glyconnect.expasy.org/ accessed on 8 February 2023) [30] from the Expasy portal of the SIB Swiss Institute of Bioinformatics. The representative structures of the 52 EGFR-related glycans were chosen. Structures of *N*-glycan models were created using a “Build via Text” tool from the GLYCAM web server (https://glycam.org accessed on 8 February 2023) [29], based on the monosaccharide sequence on the GlyConnect platform. Structures from the main conformers were selected according to differences in the position of their glycan branches and were considered for the docking process.

To test the effect of mutations on lectin-glycan docking, two rTBL mutant structures were generated using the PyMOL software version 2.5.1 [44], considering the R103Q and R130Q experimentally obtained mutants. As the available rTBL-1 sequence (PDB ID: 6tt9) has one extra alanine beside the first serine of the rTBL-1 sequence [14], it was necessary to go through the numbering one position. Therefore, only for the docking analysis, R103 changes to R104, and R130 changes to R131.

Docking analysis of *N*-glycans with rTBL-1 and its two in silico-mutated structures was performed on the LZERD platform [23] according to its requirements. A list of 500 ranked models with their respective structure files was obtained for each docking, and those with higher Ranksum values were selected. However, docking models whose interactions were located in the tetrameric zone of the quaternary structure of rTBL-1 were discarded because of steric hindrance present between monomers [11]. Also, docking results were compared with the glycosylated recombinant Tepary bean lectin structure (PDB ID: 6tt9) in order to discard those whose interactions were located in the natural glycosylation site. Interactions near the mutation sites (R104Q and R131Q) and near the CBP were preferentially selected.

Interaction residues for each docking result were analyzed using the Protein-Ligand Interaction Profiler server [45,46] (PLIP-server, https://plip-tool.biotec.tu-dresden.de/plip-web/plip/index accessed on 20 February 2023) and the software Discovery Studio Visualizer version 21.1.0.20298 (Dassault Systèmes Biovia Corp. Velizy-Villacoublay, France) [47]. In addition, the types of interactions and the solvation free energy gain of the interfaces formed in these interactions were analyzed. using the PDBePISA web server [42] (http://www.ebi.ac.uk/pdbe/prot_int/pistart.html accessed on 20 February 2023). Lectin-interacting residues of each docking result were identified and corroborated by PLIP-server [45]. Discovery Studio software [47] and the PDBePISA web server [42]. The total solvation energy gain upon formation of the interface of the interactions was obtained as the sum of the solvation energy gain corresponding to each of the protein-glycan interacting residues, expressed in kCal/mol, upon formation of the interface (Appendix A).

## 4. Conclusions

The analysis of lectin sequences revealed that the *P. acutifolius* lectin, and consequently rTBL-1 have arginine residues at positions 103 and 130 (104 and 131 for docking analysis, respectively). The conservative substitution of either of these residues by glutamine in recombinant rTBL-1 mutants led to significantly decreased cytotoxicity towards colon cancer cells, suggesting that both residues are important for the establishment of lectin-*N*-glycan interactions. Moreover, R130Q inhibited cytotoxicity towards HT-29 human colon cancer cells by two-fold more than R103Q, which is indicative of distinct roles for each of the arginine residues.

Docking analysis for previously studied experimental *N*-glycans recognized by rTBL-1, showed correlation for the interaction in the CBP (close to R104 and R131). Just two of the experimentally recognized *N*-glycans interacted with the CBP, and these were highly sialylated and core-fucosylated *N*-glycans. The same pattern was observed for 32 of 52 EGFR-related *N*-glycans, where the low molecular mass (average mass: 1.77 ± 0.33 kDa) bisected *N*-glycans have no interaction at all, and 18 EGFR-related *N*-glycans with an average mass of 2.26 ± 0.25 kDa had similar interaction zones with rTBL-1 without interacting with CBP, such as those shown by the experimentally recognized *N*-glycans. Only the high molecular mass *N*-glycans (average mass: 4.05 ± 0.25 kDa) with sialylated four antennae and core-fucosylated structure had a direct docking interaction in the CBP of rTBL-1.

The interaction between the recombinant lectin and certain EGFR *N*-glycans occurs within a complex biological context, where the final effect depends on multifactorial parameters. However, this mutant study provides a better understanding of the molecular mechanism of action. It is also likely that EGFR is one of the target molecules for the rTBL-1, and many other membrane glycoproteins could be interacting, as aberrant sialylation and fucosylation are common in other glycoproteins involved in the development of many types of cancers, including colon cancer.

This research has expanded our understanding of how the Tepary bean lectin affects the microenvironment within which cancerous cells develop, creating new opportunities to uncover the mechanisms by which this lectin exerts its anticancer effects. Further studies are needed to gain a deeper understanding of the molecular interactions related to the lectin’s cytotoxic effects to develop new strategies against EGFR-dependent cancers, particularly colon cancer.

## Figures and Tables

**Figure 1 ijms-25-13258-f001:**
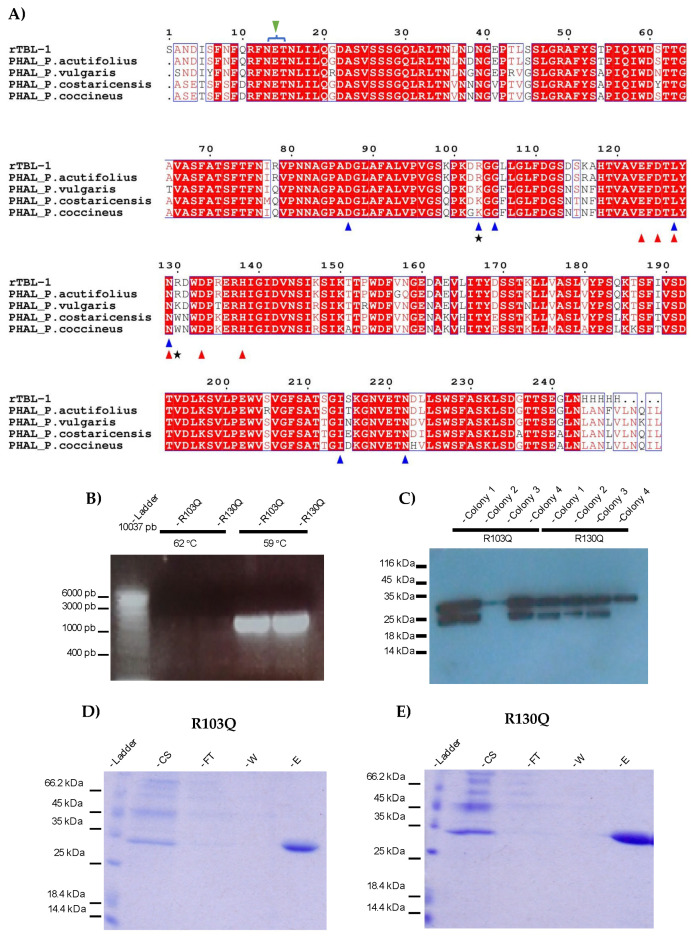
Mutants’ design and production. (**A**) Sequence alignment of rTBL-1 homologous lectins. PHAL_*Phaseolus costaricensis* (Q5ZF34_9FABA), PHAL_*Phaseolus coccineus* (Q84RP8_PHACN), PHAL_*Phaseolus vulgaris* (PHAL_PHAVU), PHAL_*Phaseolus acutifolius* (Q40750_PHAAT), rTBL-1 sequence [14]. White letters represent identical residues, semiconserved residues are displayed in red enclosed into empty boxes and non-conserved residues are depicted in black. Blue triangles show residues that correspond to the CBP similarly to PHA-L from *P. Vulgaris*; red triangles show residues interacting with metal cations; green triangle depicts sequence for *N*-glycosylation marked with a blue bracket; black stars show residues presumably responsible for the cytotoxic effect of rTBL-1 (R103 and R130). (**B**) Agarose gel electrophoresis showing pGAPαZB-rTBL-1 vector amplification. Lane 1, ladder; lanes 2–5, amplification products. (**C**) Western blot screening using anti-His antibodies for transformed colonies. (**D**,**E**) R103Q and R130Q SDS-PAGE electrophoretic profiles stained for total protein. Lane 1, ladder; lane 2, 25 µL of culture supernatant (CS); lane 3, 25 µL of flow through (FT); lane 4, 25 µL wash with 10 mM imidazole buffer (W); and lane 5, 25 µL elution with 200 mM imidazole buffer (**E**). PHAL, Leukocyte phytohemagglutinin.

**Figure 5 ijms-25-13258-f005:**
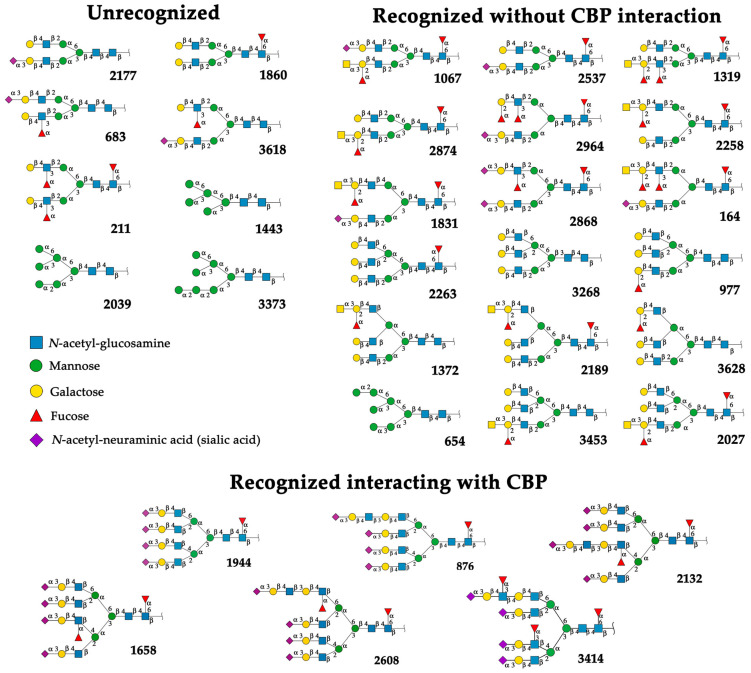
EGFR-related *N*-glycans used in docking process with rTBL-1. Graphical 2D representation of the *N*-glycans was created according to SNFG [28]. All EGFR-related *N*-glycans were named according to the GlyConnect platform [30]. CBP, carbohydrate-binding pocket.

**Table 1 ijms-25-13258-t001:** Comparison of the solvation free energy gain upon formation of the interface (Δ^i^G) of the complex EGFR-related *N*-glycans with the rTBL-1 and mutated Tepary bean lectins (R104Q, and R131Q).

2D-Graphical *N*-glycan Structure According to (SNFG) [28]	*N*-glycan ID [30]	Reported Molecular Mass (kDa) [30]	Δ^i^G (Kcal/mol)
rTBL-1	R104Q	R131Q
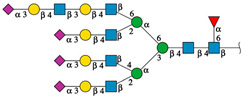	876	4.05			
−12.79	−6.57	−8.03
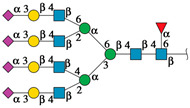	1944	3.68			
−7.99	−6.44	−8.20
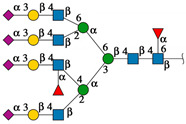	1658	3.83			
−10.39	−13.80	−15.09
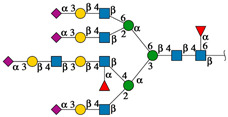	2132	4.19			
−10.99	−3.18	−11.75
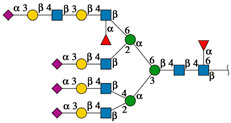	2608	4.19			
−7.86	−7.50	−10.99
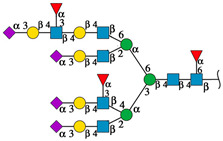	3414	4.34			
−12.59	−8.91	−5.83
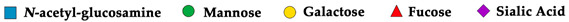

rTBL-1, Recombinant lectin; R104Q, Recombinant R104Q mutant lectin; R131Q, Recombinant R131Q mutant lectin.

## Data Availability

Data of mutants generation can be found in the repository link https://repositorio.cinvestav.mx/handle/cinvestav/1634.

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
