# Peer review of "Directed Mutagenesis for Arginine Substitution of a Phaseolus acutifolius Recombinant Lectin Disrupts Its Cytotoxic Activity"

_ijms, 2024, doi:10.3390/ijms252413258_

Round 1
Reviewer 1 Report
Comments and Suggestions for Authors
The manuscript sent for review is very interesting and has valuable results.
The work focused on determining the contribution of two arginine residues, R103 and R130, towards cytotoxic activity. Two rTBL-1 mutants were produced in Pichia pastoris, biochemically determined, and cytotoxic effects were evaluated on human colorectal cancer cells (HT-29). The Authors showed that substituting either of the arginine residues with glutamines resulted in significant losses of cytotoxic activity two and three times for R103 and R130, respectively. Docking analysis showed that the mutations decreased lectin affinity binding to some EGFR-related N-glycans. Together, these findings confirm that both of the selected arginine residues (R103 and R130) play a key role in the recognition of tumor cell glycoconjugates by rTBL-1.
I have only a minor editorial note: in several places in the text a strange symbol appears in places determining the concentration.
Author Response
Thank you very much for your comments. The symbols were replaced. We have reviewed the whole document in order to improve it.
Reviewer 2 Report
Comments and Suggestions for Authors
Dear Editor,
The manuscript by Teresa García-Gasca et al. describes the characterization and cytotoxic effects of rTBL lectins. The study suggests that arginine residues are crucial for the cytotoxic activities and Lectin-N-glycan interactions in colon cancer models. In addition, docking analysis identified a EGFR-related N-glycan suggesting a similar correlation between arginine residues and activities. This study opens new avenues about the interactions mechanisms of lectins in cancer.
The manuscript is well structured, designed and all the material and method adequately described. In think that this piece of work is suitable for publishing on Int. J. Mol. Sci. as it is.
Author Response
Thank you very much for your comments. We have reviewed the whole document in order to improve itReviewer 3 Report
Comments and Suggestions for Authors
Comments on the manuscript by Dania et al.
This manuscript by Dania et al. reports on the role of a pair of arginine amino-acid residues in the cytotoxicity of recombinant Tepary bean lectin. Authors present some very sound findings, but the manuscript requires some changes and justification to improve readability. I have the following major comments:
1. How the authors choose the criterion for determining when a particular glycan interacts or is recognized by lectin in docking studies. For example, GA-01 and GA-02 also show binding to lectin in Fig. 4 but are not counted as being recognized. The authors need to clarify this.
2. In Fig. 6, it is unclear what the difference is between the figures on the left and right for each glycan. Is it only the orientation and depth cue? They should mention this in figure legend.
3. In Table 1, how was the dissociation free energy calculated? Are say referring the docking scores are dissociation energy? What are the units of energy?
4. The authors appear to be calculating binding energy or docking results and solvation energy. These values should be reported, and the authors should comment on whether either of these energies correlates well with their binding interpretation.
5. In line 294, the term "ΔiG" is used. What does "i" represent?
Author Response
This manuscript by Dania et al. reports on the role of a pair of arginine amino-acid residues in the cytotoxicity of recombinant Tepary bean lectin. Authors present some very sound findings, but the manuscript requires some changes and justification to improve readability. I have the following major comments:
- How the authors choose the criterion for determining when a particular glycan interacts or is recognized by lectin in docking studies. For example, GA-01 and GA-02 also show binding to lectin in Fig. 4 but are not counted as being recognized. The authors need to clarify this.
The criteria for choosing the glycans to be considered for the final analyses, which were intended to see the effect of the mutations in the CBP residues, were established based on which type of glycans were most likely to interact precisely in that area. To do so, the first step was to perform a docking analysis with rTBL-1 and those N-glycans experimentally recognized [13]. It was interesting that all of them had different positions on the surface of the protein, but those with highly elongated branches and with four sialylated branches, showed a large part of the glycan interacting directly with the CBP. However, even though the first two glycans (GA-01 and GA-02) also had one or two contact points in that area, they were not taken as a reference, because they didn’t have a representative area of ​​interaction, and that is why the features of N-glycans GA-05 and GA-08 were emphasized, whose interaction area was considerable in the CBP. The cancer-related glycans were subsequently analysed and only those that had a considerable interaction area with CBP were selected, in order to be able to perform the respective analysis with the in silico-mutated proteins. The modification is shown in lines 210-216.
- In Fig. 6, it is unclear what the difference is between the figures on the left and right for each glycan. Is it only the orientation and depth cue? They should mention this in figure legend.
The Figure 6 was corrected according to the observations given and the explanation was modified according to this change. The right images corresponded to the surface interactions between N-glycans and rTBL-1 from the docking results, and the left images corresponded to the residue numbers of the residues of those interactions. However, the corrected Figure 6 will show only the docking results to simplify the descriptions of the results and their subsequent discussion and interpretation. The other images correspond to tridimensional interaction maps and were separated in a new Figure (Figure 7).
- In Table 1, how was the dissociation free energy calculated? Are say referring the docking scores are dissociation energy? What are the units of energy?
One of the advantages of using the LZERD web server to perform docking analyses is the flexibility with which this kind of analyses can be performed with very large ligands, in this case N-glycans, which have multiple interaction sites. However, using this procedure it is not possible to obtain a measure of the binding free energy (ΔG) of such interaction sites, given the large number of probable results and their subsequent analysis. For this reason, the arguments for the description of binding energies were replaced by careful analysis of the real meaning and discussed as part of the description of the docking analysis. The final results were given by the interaction interface analysis procedure on the PDBsPISA website [42] of the solvation free energy gain upon the formation of interaction interfaces (ΔiG, expressed in Kcal/mol). The ΔiG parameter was calculated as a difference in solvation energies of all residues between interfacing structures, negative values correspond to hydrophobic interfaces, indicating positive protein affinity. Therefore, corrections were made in the interpretation of this analysis for validation of the residues participating in each one of the interactions, replacing the discussions based on binding free energy. The explanation has been added in lines 355-393, including Table 1.
- The authors appear to be calculating binding energy or docking results and solvation energy. These values should be reported, and the authors should comment on whether either of these energies correlates well with their binding interpretation.
We appreciate the observation related to free binding energy reported on this paper. The docking analysis using LZERD web server can be performed with very large ligands, unlike other docking procedures. Nevertheless, under this procedure it is not possible to obtain a free binding energy of the resulting interactions and that is why the discussion on free binding energies was corrected. The term (ΔiG) indicated the solvation free energy gain upon formation of the interface, expressed in Kcal/mol and it is calculated as difference in solvation energies of all residues between interfacing structures, in this case between each N-glycan and the lectin. Negative values of ΔiG corresponds to hydrophobic interfaces (positive protein affinity). The PDBePISA webserver [42] allowed us the calculation of ΔiG changes for all possible interactions in docking results, including those related to the interaction between lectin monomers or between glycan monosaccharides. By the nature of the analysis, it was necessary to isolate and focus only on the N-glycan-protein interactions. For the analysis of the interface interactions two webservers and one program to validate them were used.
Due to the numerous points of interaction of complex N-glycans on the surface of the lectin, the calculation of a standard measure of free binding affinity was not possible, the discussion based on free binding energies was replaced and the ΔiG changes as indicators of positive protein affinity interactions between the lectins and N-glycans was included. The explanation has been added in lines 355-393, including Table 1.
- In line 294, the term "ΔiG" is used. What does "i" represent?
The term (ΔiG) indicates the solvation free energy gain upon formation of the interface. The “i” represents solvation free energy gain.
Thank you very much for your comments, they were very important in order to improve the manuscript.
Reviewer 4 Report
Comments and Suggestions for Authors
In this work Martínez-Alarcónis and co-workers compared a recombinant lectin (rTBL-1), known for its cytotoxic effect against colon cancer cells, with two mutants of this protein, to evaluate the contribution of two particular arginine residues in its activity. The work includes an experimental and a computational section, and is overall well-written, but I have some major concerns that the authors should address.
Comments on the experimental section:
- Lines 180-186: it is not clear why the authors state that residue R103 is more important in the cytotoxic effect than R130, if mutation R130Q results in a higher LC50 and a higher reduction in its cytotoxic effect as shown in Figure 3.
- Figure 3: it is not clear to me what is the significance of each small letter and how they show significative differences for each separate treatment at different concentrations, as stated in the legend.
Comments on the computational section:
- Figure 4: is not clear which protein and docking is represented in the image. Are the dockings of the three lectins superposed in one image? How can we distinguish the poses resulting from the three docking simulations? Also, no orange sticks is in the figure, unlike what stated in the legend.
- Paragraph 2.4.2: Specify in the text (and in the methods) if these dockings were also performed using LZERD and with the same parameters of the following section.
- Figure 5: I think the phrase "the N-glycans from the mammalian glycan array “Version 5.4” are labeled as GA-## where ## is the number described by Martínez-Alarcón et al. (2020) and is related to the signal’s relative intensity of lectin recognition. " should belong to the previous figure since there is no “GA” in this image.
- Figure 6: the labels are too small and impossible to read, and all images are very blurry. Either increase the size of labels or, maybe better, the size of the images by arranging the panels differently.
- Paragraph 2.4.3: in my opinion, this part should be rewritten in a clearer (and perhaps) schematic manner, because it was difficult to follow what the authors were trying to describe, and the comparisons they were making among the docking results of differently branched glycans. This is also because it was hard to compare the individual results obtained for the different glycans and/or lectins reported in the Supplementary Material tables. While it is useful to have the tables with complete results for reference, I suggest the authors to find a way to gather the most important results from the Supplementary Materials tables, to better highlight the comparison they’re discussing and facilitate a clearer understanding of paragraph 2.4.3.
Finally, the conclusions drawn from these results are not clear. I think overall the computational results suggest the mutation effect is very dependent on the glycan type (in some cases the mutated residue establishes stronger contacts with the glycans than the wt residue, in other cases energy differences are observed without the interaction of wt / mutated residues). Therefore, could the cytotoxic effect of the recombinant and/or mutated lectins be dependent on the specific glycans exposed on targeted cells?
Also, higher differences in energies and interaction patterns were detected in the R104Q docking simulations, suggesting a decreased interaction of this mutant with glycans and thus a reduced effect on cancer cells. This is different from the outcome of experimental results, where R140Q cytotoxic effect is evidently lower than that of R103Q. The authors should discuss this and reason on this apparent discrepancy.
Minor corrections:
- Some references are indicated by superscript numerals instead of bracketed numbers
- The Supplementary Materials look a bit stretched and blurred
Author Response
In this work Martínez-Alarcón is and co-workers compared a recombinant lectin (rTBL-1), known for its cytotoxic effect against colon cancer cells, with two mutants of this protein, to evaluate the contribution of two particular arginine residues in its activity. The work includes an experimental and a computational section, and is overall well-written, but I have some major concerns that the authors should address.
Comments on the experimental section:
Lines 180-186: it is not clear why the authors state that residue R103 is more important in the cytotoxic effect than R130, if mutation R130Q results in a higher LC50 and a higher reduction in its cytotoxic effect as shown in Figure 3.
Thank you for your comment. You are right, a higher inhibition of cytotoxicity is observed with R130Q, we have corrected the text, lines 184-189.
Figure 3: it is not clear to me what is the significance of each small letter and how they show significative differences for each separate treatment at different concentrations, as stated in the legend.
Thank you for your comment. We compared each treatment against the control cells (no added protein). We have indicated the specific small letters for each treatment, being letters a-e for rTBL-1, f-h for R103Q and letter i for R130Q, lines 194-195. Also, a statement was included in Materials and Methods, lines 510-512.
Comments on the computational section:
Figure 4: is not clear which protein and docking is represented in the image. Are the dockings of the three lectins superposed in one image? How can we distinguish the poses resulting from the three docking simulations? Also, no orange sticks is in the figure, unlike what stated in the legend.
Figure 4 shows docking only for rTBL-1 protein, the mutants were not used at this point because we only used experimental glycans that were found to interact with rTBL-1 (Martinez-Alarcon et al 2020). This experiment identified which glycans exhibited stronger interactions enabling us to search for EGFR glycans in the data base.
Paragraph 2.4.2: Specify in the text (and in the methods) if these dockings were also performed using LZERD and with the same parameters of the following section.
All docking analysis was performed using the LZERD web server (lines 201-203) with all specifications described in the Materials and Methods section, paragraph 3.7.
Figure 5: I think the phrase "the N-glycans from the mammalian glycan array “Version 5.4” are labeled as GA-## where ## is the number described by Martínez-Alarcón et al. (2020) and is related to the signal’s relative intensity of lectin recognition. " should belong to the previous figure since there is no “GA” in this image.
Thank you, you are right. The legends of Figures 4 and 5 were modified.
Figure 6: the labels are too small and impossible to read, and all images are very blurry. Either increase the size of labels or, maybe better, the size of the images by arranging the panels differently.
Figure 6 was corrected in order to highlight the topological representation of docking analysis results, and the labels of the structures. Interaction maps were described in a new Figure (Figure 7).
Paragraph 2.4.3: in my opinion, this part should be rewritten in a clearer (and perhaps) schematic manner, because it was difficult to follow what the authors were trying to describe, and the comparisons they were making among the docking results of differently branched glycans. This is also because it was hard to compare the individual results obtained for the different glycans and/or lectins reported in the Supplementary Material tables. While it is useful to have the tables with complete results for reference, I suggest the authors to find a way to gather the most important results from the Supplementary Materials tables, to better highlight the comparison they’re discussing and facilitate a clearer understanding of paragraph 2.4.3.
We appreciate the observations; the information was reorganized according to the main findings and the relevance of them. Figure 6 was corrected and a Figure 7 was added in order to improve the clarity and relevance of the results. Also, the supplementary material was corrected and a schematic explanation of the proposed nomenclature for complex N-glycans was attached. The main discussion was reorganized highlighting the main points and simplifying the comparisons based on the new organization of figures. Docking results and their subsequent discussion were organized as follows: docking interaction positions, interaction residues in each docking interaction result and the discussion using the solvation free energy gain upon formation of the interface as parameter of affinity gaining between the interfaces of the interactions.
Finally, the conclusions drawn from these results are not clear. I think overall the computational results suggest the mutation effect is very dependent on the glycan type (in some cases the mutated residue establishes stronger contacts with the glycans than the wt residue, in other cases energy differences are observed without the interaction of wt / mutated residues). Therefore, could the cytotoxic effect of the recombinant and/or mutated lectins be dependent on the specific glycans exposed on targeted cells?
Also, higher differences in energies and interaction patterns were detected in the R104Q docking simulations, suggesting a decreased interaction of this mutant with glycans and thus a reduced effect on cancer cells. This is different from the outcome of experimental results, where R140Q cytotoxic effect is evidently lower than that of R103Q. The authors should discuss this and reason on this apparent discrepancy.
Thank you, we have reorganized the discussions and conclusions after reviewing all the comments and correcting the text, particularly in lines 394-406 and in the conclusions section.
Minor corrections:
Some references are indicated by superscript numerals instead of bracketed numbers
Thank you, all the references were reviewed and corrected.
The Supplementary Materials look a bit stretched and blurred
The letter type of all the supplementary material was changed.
Thank you very much for your comments, they were very important in order to improve the manuscript.
Round 2
Reviewer 4 Report
Comments and Suggestions for Authors
The authors addressed all my comments and significantly improved the manuscript, which I believe is now suitable for publication.
Author Response
Answers to reviewer:
[Major concerns]
- Abbreviations: The use of abbreviations when writing a paper has many advantages besides simplicity of expression. To use an abbreviation, first write the abbreviation in parentheses after the full name, and then use the abbreviation from Introduction to the final Conclusion. Abbreviations should only be used if they are repeatedly used and if they are not used again, only the full name should be used. In particular, because of the characteristics of IJMS, where Materials and Methods is arranged at the end of the paper, the original words and abbreviations are written in the order they are used from the introduction, and only when the abbreviation is used repeatedly, the abbreviation can be used until the conclusion.
Thank you, the document was revised in order to assure that all the abbreviations were well written.
2. In cases where abbreviations are used within figures or tables, please list these abbreviations along with their corresponding full names in the figure legends or at the bottom of corresponding tables. If there are two or more abbreviations, arrange them in alphabetical order.
Thank you we revised all figures and the table in order to include the abbreviations at the end of the figures legends or at the bottom of the table.
3. Materials and Methods section - When naming a particular chemical company, you must provide location information such as company name, city and/or state (abbreviation in the USA and Canada) and country. Once you have named a company with the information, you should only mention a company’s name thereafter.
Thank you we have revised all the company names and for those that were mentioned several times the city and country were omitted.
[Minor concerns]
1. Line 27: Define EGFR in the abstract. OK
2. Line 39: Define PHA. OK
3. Line 48: Abbreviate P. pastoris here and use it later. Here we wrote the full name as is the first time that is used in the introduction. Later is abbreviated in the whole text.
4. Line 50: Define TBLF. OK
5. Line 52: Define TBL-1. OK
6. Line 71: Define Ig. OK
7. Lines 142 and 143: Write the scientific names of plants with spaces. OK
8. Lines 158 and 162: ‘O’ and ‘N’ should be written in italics. There are several similar typos. Thank you, all the wrong typos were corrected.
9. Lines 169, 170, 171, and 172: The names of common compounds are not typically written with the first letter capitalized, so please change them all to lowercase. Also, arranging them in the required alphabetical order will make it easier to understand. Ok, we tried to do the best
10. Line 418: Define E. coli. If not used repeatedly, there is no need to abbreviate E. coli. OK
11. Line 434: Define YPG. OK
12. Lines, 438, 441, and 501: Two forms of time units are used, but in my opinion, just 'h' would be sufficient. We found only “h”, we agree using it as hour abbreviation.
13. Line 469: The speed of the centrifuge should be expressed in gravity (g) rather than rpm. When indicating centrifugal force, either italicize the 'g' representing gravity or write it as 'x g'. Ok
Overall, the manuscript can be considered to publication after minor revision as indicated above.
Thank you very much, your observations were very important in order to improve the manuscript.
